# Conditional Meta-Learning of Linear Representations

**Giulia Denevi**[*]
Leonardo Labs (Italy)
giulia.denevi.ext@leonardo.com

**Massimiliano Pontil**
Istituto Italiano di Tecnologia (Italy) & University College of London (UK)
massimiliano.pontil@iit.it

**Carlo Ciliberto**
University College of London (UK) & Istituto Italiano di Tecnologia (Italy)
c.ciliberto@ucl.ac.uk

## Abstract

Standard meta-learning for representation learning aims to find a common representation to be shared across multiple tasks. The effectiveness of these methods is often limited when the nuances of the tasks' distribution cannot be captured by a single representation. In this work we overcome this issue by inferring a conditioning function, mapping the tasks' side information (such as the tasks' training dataset itself) into a representation tailored to the task at hand. We study environments in which our conditional strategy outperforms standard meta-learning, such as those in which tasks can be organized in separate clusters according to the representation they share. We then propose a meta-algorithm capable of leveraging this advantage in practice. In the unconditional setting, our method yields a new estimator enjoying faster learning rates and requiring less hyper-parameters to tune than current state-of-the-art methods. Our results are supported by preliminary experiments.

## 1 Introduction

Learning a shared representation among a class of machine learning problems is a well-established approach used both in multi-task learning [3, 20, 11] and meta-learning [18, 15, 5, 17, 35, 24, 30, 9, 7]. The idea behind this methodology is to consider two nested problem: at the within-task level an empirical risk minimization is performed on each task, using inputs transformed by the current representation, on the outer-task (meta-) level, such a representation is updated taking into account the errors of the within-task algorithm on previous tasks.

Such a technique was shown to be advantageous in contrast to solving each task independently when the tasks share a low dimensional representation, see e.g. [27, 25, 15, 24, 35, 5, 22, 9]. However, in real world applications we often deal with heterogeneous classes of learning tasks, which may overall be only loosely related. Consequently, the tasks' commonalities may not be captured well by a single representation shared among all the tasks. This is for instance the case in which the tasks can be organized in different groups (clusters), where only tasks belonging to the same cluster share the same low-dimensional representation.

In order to overcome this issue, previous authors developed non-convex methods (or convex relaxations) attempting at clustering the tasks, see e.g. [4, 26, 2, 20, 28, 40, 38, 31]. In this work, we follow

---

[*]Work done while the first author was with Istituto Italiano di Tecnologia (Italy).

36th Conference on Neural Information Processing Systems (NeurIPS 2022).

the recent literature on heterogeneous meta-learning [37, 36, 32, 21, 10, 39, 14, 7] and propose a so-called *conditional meta-learning* approach for meta-learning a representation. Our algorithm learns a conditioning function mapping available tasks' side information into a *linear* representation that is tuned to that task at hand. Our approach borrows from [14], where the authors proposed a conditional meta-learning approach for fine tuning and biased regularization. In those cases however, the tasks' target vectors are assumed to be all close to a common bias vector rather than sharing the same low-dimensional linear representation, as instead explored in this work. As we explain in the following, working with the representation setting requires a significant contribution with respect to the bias one in order to give a new formulation of the problem, to consider a different meta-objective and a different interpretation of the results. In addition, the representation setting is known to be a more relevant and effective framework in many scenarios in comparison to the bias one (see e.g. [28]).

In this work, we propose for the first time an online conditional method for linear representation learning with strong theoretical guarantees. In particular, we show that the method is advantageous over standard (unconditional) representation learning methods used in meta-learning when the environment of observed tasks is heterogeneous.

**Contributions and organization.** The contributions of this work are the following. First, in Sec. 2, we design a conditional meta-learning approach to infer a linear representation that is tuned to the task at hand. Second, in Sec. 3, we formally characterize circumstances under which our conditional framework brings advantage with respect to the standard unconditional approach. In particular, we argue that this is the case when the tasks are organized in different clusters according to the support pattern or linear representation their target vectors' share. Third, in Sec. 4, we design a convex meta-algorithm providing a comparable gain as the number of the tasks it observes increases. In the unconditional setting, the proposed method is able to recover faster rates and it requires to tune one less hyper-parameter with respect to the state-of-the-art unconditional methods. Finally, in Sec. 5, we present numerical experiments supporting our theoretical claims. We conclude our work in Sec. 6 and we postpone the missing proofs to the supplementary material.

## 2 Conditional representation learning

In this section we introduce our conditional meta-learning setting for representation learning. Then, we proceed to identify the differences with respect to (with respect to) the standard unconditional counterpart. We begin our overview by first introducing the class of inner learning algorithms we use in this work.

**Within-task algorithms.** We consider the standard linear supervised learning setting over $\mathcal{Z} = \mathcal{X} \times \mathcal{Y}$ with $\mathcal{X} \subseteq \mathbb{R}^d$ and $\mathcal{Y} \subseteq \mathbb{R}$ input and output spaces, respectively. We denote by $\mathcal{P}(\mathcal{Z})$ the set of probability distributions (tasks) over $\mathcal{Z}$. For any task $\mu \in \mathcal{P}(\mathcal{Z})$ and a given loss function $\ell : \mathbb{R} \times \mathbb{R} \to \mathbb{R}$, we aim at finding a weight vector $w_\mu \in \mathbb{R}^d$ minimizing the *expected risk*

$$\min_{w \in \mathbb{R}^d} \mathcal{R}_\mu(w) \qquad \mathcal{R}_\mu(w) = \mathbb{E}_{(x,y)\sim\mu} \, \ell\big(\langle x, w \rangle, y\big), \tag{1}$$

where, $\langle \cdot, \cdot \rangle$ represents the Euclidean product in $\mathbb{R}^d$. In practice, $\mu$ is only partially observed through a dataset $Z = (x_i, y_i)_{i=1}^n \sim \mu^n$, namely, a collection of $n$ identically independently distributed (i.i.d.) points sampled from $\mu$. Thus, the goal becomes to use a learning algorithm in order to estimate a candidate weight vector with a small expected risk converging to the ideal $\mathcal{R}_\mu(w_\mu)$ as the sample size $n$ grows. Specifically, in this work we will consider as candidate estimators, the family of regularized empirical risk minimizers for linear feature learning [3]. Formally, denoting by $\mathcal{D} = \bigcup_{n \in \mathbb{N}} \mathcal{Z}^n$ the space of all datasets on $\mathcal{Z}$, for a given $\theta \in \Theta$ in $\Theta = \mathbb{S}_+^d$ the set of positive definite $d \times d$ matrices, we will consider the following learning algorithms $A(\theta, \cdot) : \mathcal{D} \to \mathbb{R}^d$:

$$A(\theta, Z) = \operatorname*{argmin}_{w \in \mathrm{Ran}(\theta) \subset \mathbb{R}^d} \mathcal{R}_{Z,\theta}(w), \qquad R_{Z,\theta}(w) = \frac{1}{n} \sum_{i=1}^n \ell(\langle x_i, w \rangle, y_i) + \frac{1}{2} \langle w, \theta^\dagger w \rangle \tag{2}$$

where $\mathrm{Ran}(\theta)$ denotes the range of $\theta$. Here $\theta^\dagger$ denotes the pseudoinverse of $\theta$. Throughout this work we will denote by $\mathcal{R}_Z(\cdot) = 1/n \sum_{i=1}^n \ell(\langle x_i, \cdot \rangle, y_i)$ the empirical risk associated to $Z$. Here, $\theta$ plays the role of a linear feature representation that is learned during the meta-learning process (see [3]).

**Remark 1** (Within-task regularization parameter). *Differently to previous work, see e.g. [15], we do not impose any constraints on the trace of $\theta$ (e.g. $\text{Tr}(\theta) \leq 1$). This allows us to absorb the regularization parameter $\lambda$ typically used to control $\lambda \langle w, \theta^\dagger w \rangle$ in $\theta$. As we will discuss later, this choice reduces the number of hyper-parameter to tune and it allows to enjoy faster learning rates.*

**Remark 2** (Online variant of Eq. (2)). *Paying additional negligible logarithmic factors, our analysis and results extend also to the setting in which the minimizer in Eq. (2) is replaced by a pre-conditioned variant of online gradient descent on $\mathcal{R}_{Z,\theta}$ with starting point $w_0 = 0$ and appropriate step size:*

$$A(\theta, Z) = \frac{1}{n} \sum_{i=1}^{n} w_i, \quad w_{i+1} = w_i - \frac{\theta p_i}{i}, \quad p_i = s_i x_i + \theta^\dagger w_i, \quad s_i \in \partial \ell(\cdot, y_i)(\langle x_i, w_i \rangle). \quad (3)$$

**Unconditional Meta-Learning.** The standard unconditional meta-learning setting assumes there exist a meta-distribution $\rho \in \mathcal{P}(\mathcal{M})$ – also called *environment* in [6] – over a family $\mathcal{M} \subseteq \mathcal{P}(\mathcal{Z})$ of distributions (tasks) $\mu$ and it aims at selecting an inner algorithm in the family above that is well suited to solve tasks $\mu$ sampled from $\rho$. This target can be reformulated as finding a linear representation $\theta_\rho \in \Theta$ such that the corresponding algorithm $A(\theta_\rho, \cdot)$ minimizes the *transfer risk*

$$\min_{\theta \in \Theta} \mathcal{E}_\rho(\theta), \qquad \mathcal{E}_\rho(\theta) = \mathbb{E}_{\mu \sim \rho} \, \mathbb{E}_{Z \sim \mu^n} \, \mathcal{R}_\mu\big(A(\theta, Z)\big). \quad (4)$$

In practice, this stochastic problem is usually tackled by iteratively sampling a task $\mu \sim \rho$ and a corresponding dataset $Z \sim \mu^n$, and, then, performing a step of stochastic gradient descent on an empirical approximation of Eq. (4) computed from $Z$. This approach has proven effective for instance when the tasks of the environment share a simple common linear representation, see e.g. [18, 5, 22, 15, 16, 12, 17, 9]. However, when a single linear representation is not sufficient for the entire environment of tasks (e.g. multi-clusters), this homogeneous approach is expected to fail. In order to overcome this limitation, some recent works have adopted the following conditional approach to the problem, see e.g. [37, 36, 32, 21, 10, 39, 14].

**Conditional Meta-learning.** Analogously to [14], we assume that any task $\mu \sim \rho$ is provided of additional side information $s \in \mathcal{S}$. In such a case, we consider the environment $\rho$ as a distribution $\rho \in \mathcal{P}(\mathcal{M}, \mathcal{S})$ over the set $\mathcal{M}$ of tasks and the set $\mathcal{S}$ of possible side information. Moreover, as usual, we assume $\rho$ to decompose in $\rho(\cdot|s)\rho_\mathcal{S}(\cdot)$ and $\rho(\cdot|\mu)\rho_\mathcal{M}(\cdot)$ the conditional and marginal distributions with respect to $\mathcal{S}$ and $\mathcal{M}$. For instance, we observe that the side information $s$ could contain descriptive features of the associated task, for example attributes in collaborative filtering [1], or additional information about the users in recommendation systems [19]). Moreover $s$ could be formed by a portion of the dataset sampled from $\mu$ (see [37, 14]). Conditional meta-learning leverages this additional side information in order to adapt (or condition) the linear representation $\theta \in \Theta$ on the associated task at hand, by learning a linear-representation-valued function $\tau$ solving the problem

$$\min_{\tau \in \mathcal{T}} \mathcal{E}_\rho(\tau), \qquad \mathcal{E}_\rho(\tau) = \mathbb{E}_{(\mu,s) \sim \rho} \mathbb{E}_{Z \sim \mu^n} \mathcal{R}_\mu(A(\tau(s), Z)) \quad (5)$$

over the space $\mathcal{T}$ of measurable functions $\tau : \mathcal{S} \to \Theta$. Notice that we retrieve the unconditional meta-learning problem in Eq. (4) if we restrict Eq. (5) to the set of functions $\mathcal{T}^{\text{const}} = \{\tau \mid \tau(\cdot) \equiv \theta, \; \theta \in \Theta\}$, mapping all the side information into the same constant linear representation.

In the next section, we will investigate the theoretical advantages of adopting such a conditional perspective and, then, we will introduce a convex meta-algorithm to tackle Eq. (5).

## 3 The advantage of conditional representation learning

In order to characterize the behavior of the optimal solution of Eq. (5) and to investigate the potential advantage of conditional meta-learning, we analyze the generalization properties of a given conditioning function $\tau$. Formally, we compare the error $\mathcal{E}_\rho(\tau)$ with respect to the optimal minimum risk

$$\mathcal{E}_\rho^* = \mathbb{E}_{\mu \sim \rho} \, \mathcal{R}_\mu(w_\mu) \qquad w_\mu = \underset{w \in \mathbb{R}^d}{\operatorname{argmin}} \, \mathcal{R}_\mu(w). \quad (6)$$

In order to do this, we first need to introduce the following standard assumptions used also in previous literature. Throughout this work we will denote by $\cdot^\top$ the standard transposition operation.

**Assumption 1.** *Let $\ell$ be a convex and $L$-Lipschitz loss function in the first argument. Additionally, there exist $R > 0$ such that $\|x\| \leq R$ for any $x \in \mathcal{X}$.*

**Theorem 1** (Excess risk with generic conditioning function $\tau$). *Let Asm. 1 hold. For any $s \sim \rho_\mathcal{S}$, introduce the conditional covariance matrices $W(s) = \mathbb{E}_{\mu \sim \rho(\cdot|s)} w_\mu w_\mu^\top$ and $C(s) = \mathbb{E}_{\mu \sim \rho(\cdot|s)} \mathbb{E}_{x \sim \eta_\mu} xx^\top$, where, $\eta_\mu$ denotes the inputs' marginal distribution of the task $\mu$. Let $\tau \in \mathcal{T}$ such that $\mathrm{Ran}(W(s)) \subseteq \mathrm{Ran}(\tau(s))$ for any $s \sim \rho_\mathcal{S}$ and let $A(\tau(s), \cdot)$ be the associated inner algorithm from Eq. (2). Then,*

$$\mathcal{E}_\rho(\tau) - \mathcal{E}_\rho^* \leq \frac{\mathbb{E}_{s \sim \rho_\mathcal{S}} \mathrm{Tr}\big(\tau(s)^\dagger W(s)\big)}{2} + \frac{2L^2 \mathbb{E}_{s \sim \rho_\mathcal{S}} \mathrm{Tr}\big(\tau(s) C(s)\big)}{n}. \tag{7}$$

*Proof.* For any $(\mu, s) \sim \rho$, consider the decomposition $\mathcal{E}_\rho(\tau) - \mathcal{E}_\rho^* = \mathbb{E}_{(\mu,s) \sim \rho}\big[\mathrm{B}_{\mu,s} + \mathrm{C}_{\mu,s}\big]$, with

$$\mathrm{B}_{\mu,s} = \mathbb{E}_{Z \sim \mu^n}\Big[\mathcal{R}_\mu(A(\tau(s), Z)) - \mathcal{R}_Z(A(\tau(s), Z))\Big]$$

$$\mathrm{C}_{\mu,s} = \mathbb{E}_{Z \sim \mu^n}\Big[\mathcal{R}_Z(A(\tau(s), Z)) - \mathcal{R}_\mu(w_\mu)\Big].$$

$\mathrm{B}_{\mu,s}$ is the generalization error of the inner algorithm $A(\tau(s), \cdot)$ on the task $\mu$. Hence, applying stability arguments (see Prop. 6 in App. A), we can write $\mathrm{B}_{\mu,s} \leq 2L^2 \mathrm{Tr}\big(\tau(s) \mathbb{E}_{x \sim \eta_\mu} xx^\top\big) n^{-1}$. Regarding the term $\mathrm{C}_{\mu,s}$, for any conditioning function $\tau$ such that $w_\mu \in \mathrm{Ran}(\tau(s))$, we can write $\mathrm{C}_{\mu,s} \leq \mathbb{E}_{Z \sim \mu^n}\Big[\mathcal{R}_{Z,\tau(s)}(w_\mu) - \mathcal{R}_\mu(w_\mu)\Big] = 2^{-1} \mathrm{Tr}\big(\tau(s)^\dagger w_\mu w_\mu^\top\big)$, where, the inequality exploits the definition of the algorithm in Eq. (2) as minimum of the regularized empirical risk. The desired statement follows by combining the two bounds above and rewriting $\mathbb{E}_{(\mu,s) \sim \rho} = \mathbb{E}_{s \sim \rho_\mathcal{S}} \mathbb{E}_{\mu \sim \rho(\cdot|s)}$. $\square$

Thm. 1 suggests that the conditioning function $\tau_\rho$ minimizing the right hand side of Eq. (7) is a good candidate to solve the meta-learning problem. The following result explores this question by showing that such a minimizer admits a closed form solution. The proof is reported in App. B. In the following, we will denote by $\|\cdot\|_F$ and $\|\cdot\|_*$ the Frobenius and trace norm of a matrix, respectively.

**Proposition 2** (Best conditioning function in hindsight). *The conditioning function minimizer and the minimum of the bound presented in Thm. 1 over the set $\{\tau \in \mathcal{T} \mid \mathrm{Ran}(W(s)) \subseteq \mathrm{Ran}(\tau(s)), \ \rho_\mathcal{S}\text{-almost surely}\}$, are respectively*

$$\tau_\rho(s) = (2L)^{-1} n^{1/2} C(s)^{\dagger/2} (C(s)^{1/2} W(s) C(s)^{1/2})^{1/2} C(s)^{\dagger/2}$$

$$\mathcal{E}_\rho(\tau_\rho) - \mathcal{E}_\rho^* \leq 2L \mathbb{E}_{s \sim \rho_\mathcal{S}} \big\| W(s)^{1/2} C(s)^{1/2} \big\|_* n^{-1/2}. \tag{8}$$

We observe that, in comparison to [14], the numerator term in the bound above describes a different kind of tasks' similarity assumption: the conditional variance term $\mathbb{E}_{(\mu,s) \sim \rho} \| w_\mu - \mathbb{E}_{\mu \sim \rho(\cdot|s)} w_\mu \|^2$ present in [14] is now substituted by the trace norm term $\mathbb{E}_{s \sim \rho_\mathcal{S}} \big\| W(s)^{1/2} C(s)^{1/2} \big\|_*$ above. Additionally, the bound above allows us to quantify the benefits of adopting the conditional feature learning strategy.

**Conditional vs. unconditional Meta-Learning.** Applying Prop. 2 to $\mathcal{T}^{\mathrm{const}}$, we obtain the optimal (constant) meta-parameter and the corresponding excess risk bound for unconditional meta-learning

$$\tau \equiv \theta_\rho = (2L)^{-1} n^{1/2} C_\rho^{\dagger/2} (C_\rho^{1/2} W_\rho C_\rho^{1/2})^{1/2} C_\rho^{\dagger/2}$$

$$\mathcal{E}_\rho(\theta_\rho) - \mathcal{E}_\rho^* \leq 2L \big\| W_\rho^{1/2} C_\rho^{1/2} \big\|_* n^{-1/2} \tag{9}$$

with unconditional covariance matrices $W_\rho = \mathbb{E}_{\mu \sim \rho} w_\mu w_\mu^\top$ and $C_\rho = \mathbb{E}_{\mu \sim \rho} \mathbb{E}_{x \sim \eta_\mu} xx^\top$. We observe that in the previous literature [13, 15] the authors restricted the unconditional problem over the smaller class of linear representation $\hat{\Theta} = \{\theta \in \mathbb{S}_+^d : \mathrm{Ran}(W_\rho) \subseteq \mathrm{Ran}(\theta), \mathrm{Tr}(\theta) \leq 1\}$ and they considered as the best unconditional representation, the matrix minimizing only a part of the previous bound, namely,

$$\hat{\theta}_\rho = \underset{\theta \in \hat{\Theta}}{\mathrm{argmin}} \, \mathrm{Tr}\big(\theta^\dagger W_\rho\big) = W_\rho^{1/2} \big(\mathrm{Tr}\big(W_\rho^{1/2}\big)\big)^{-1}. \tag{10}$$

On the other hand, the unconditional oracle we introduce above in Eq. (9) allows us to recover a tighter bound which is able to recover the best performance between independent task learning (ITL) and the oracle considered in previous literature [15]. Indeed, by exploiting the duality between the trace norm $\|\cdot\|_*$ and the operator norm $\|\cdot\|_\infty$ of a matrix, we can upper bound the right-side-term in Eq. (9) by the quantity

$$2L \min \Big\{ \big\| W_\rho^{1/2} \big\|_* \big\| C_\rho^{1/2} \big\|_\infty, \big\| W_\rho^{1/2} \big\|_F \big\| C_\rho^{1/2} \big\|_F \Big\} n^{-1/2},$$

namely, the minimum between the bound for independent task learning and the bound for unconditional oracle obtained by previous authors. Notice that the unconditional quantity in Eq. (9) is always bigger than the conditional quantity in Eq. (8), since Eq. (9) coincides with the minimum over a smaller class of function. In order to quantify the gap between these two quantities – namely, the advantage in using the conditional approach with respect to the unconditional one – we have to compare the term $\left\| W_\rho^{1/2} C_\rho^{1/2} \right\|_*$ with the term $\mathbb{E}_{s \sim \rho_{\mathcal{S}}} \left\| C(s)^{1/2} W(s)^{1/2} \right\|_*$.

We report below a setting that can be considered illustrative for many real-world scenarios in which such a gap in performance is significant.

**Example 1** (Clusters). *Let $\mathcal{S} = \mathbb{R}^q$ be the side information space, for some integer $q > 0$. Let $\rho$ be such that the side information marginal distribution $\rho_{\mathcal{S}}$ is given by a uniform mixture of $m$ uniform distributions. More precisely, let $\rho_{\mathcal{S}} = \frac{1}{m} \sum_{i=1}^m \rho_{\mathcal{S}}^{(i)}$, with $\rho_{\mathcal{S}}^{(i)} = \mathcal{U}(\mathcal{B}(a_i, 1/2))$ the uniform distribution on the ball of radius $1/2$ centered at $a_i \in \mathcal{S}$, characterizing the cluster $i$. For a given side information $s$, a task $\mu \sim \rho(\cdot|s)$ is sampled such that: 1) its inputs' marginal $\eta_\mu$ is a distribution with constant covariance matrix $C(s) = \mathbb{E}_{\mu \sim \rho(\cdot|s)} \mathbb{E}_{x \sim \eta_\mu} x x^\top = C$, for some $C \in \mathbb{S}_+^d$, 2) $w_\mu$ is sampled from a distribution with conditional covariance matrix $W(s) = \mathbb{E}_{\mu \sim \rho(\cdot|s)} w_\mu w_\mu^\top$, with $W(s)$ such that $(C^{1/2} W(s) C^{1/2})(C^{1/2} W(p) C^{1/2}) = 0$ if $s \neq p$. Then, $\mathbb{E}_{s \sim \rho_{\mathcal{S}}} \left\| C(s)^{1/2} W(s)^{1/2} \right\|_* = \frac{1}{\sqrt{m}} \left\| W_\rho^{1/2} C_\rho^{1/2} \right\|_*$.*

The inequality above tells us that, in the setting of Ex. 1, the conditional approach gains a $\sqrt{m}$ factor in comparison to the unconditional approach. Therefore, the larger the number of clusters is, the more pronounced the advantage of conditional approach with respect to the unconditional one will be. The $\sqrt{m}$ gain factor follows from the fact that the weight vectors $w_\mu$ sampled from the different clusters share disjoint supports (they share orthogonal representations). This allows us to rewrite the overall clusters weight vectors' covariance as the average of the intra clusters weight vectors' covariances. The $\sqrt{m}$ term comes from this rewriting and the quadratic behavior of the covariance matrix. We refer to App. C for more details and the deduction. We also observe that a particular case of the setting above could be that one in which $q = 1$ and the side information are *noisy* observations of the index of the cluster the tasks belong to. In our experiments, in Sec. 5, we consider a more interesting and realistic variant of the setting above, in which we will use as task's side information a training dataset sampled from that task. In the next section, we introduce a convex meta-algorithm mimicking this advantage also in practice.

## 4 Conditional representation Meta-Learning algorithm

To tackle conditional meta-learning in practice we consider a parametrization where the conditioning functions that are modeled with respect to a given feature map $\Phi : \mathcal{S} \to \mathbb{R}^k$ (with $k \in \mathbb{N}$) on the side information space. In other words, we consider $\tau : \mathcal{S} \to \mathbb{S}_+^d$,

$$\tau(\cdot) = \left( M\Phi(\cdot) \right)^\top M\Phi(\cdot) + C, \tag{11}$$

for some tensor $M \in \mathbb{R}^{p \times d \times k}$ ($p \in \mathbb{N}$) and matrix $C \in \mathbb{S}_+^d$. By construction, the above parametrization guarantees us to learn functions taking values in the set of positive semi-definite matrices. However, directly addressing the meta-learning problem poses two issues: first, dealing with tensorial structures might become computationally challenging in practice and second, such parametrization is quadratic in $M$ and would lead to a non-convex optimization functional in practice. To tackle this issue, the following results shows that we can rewrite the conditioning function in the form of Eq. (11) by using a matrix in $\mathbb{S}_+^{dk}$. This will allows us to implement our method working with matrices in $\mathbb{S}_+^{dk}$, instead of tensors in $\mathbb{R}^{p \times d \times k}$. Throughout this work, we will denote by $\otimes$ the Kronecker product.

**Proposition 3** (Matricial re-formulation of $\tau_M(s)$). *Let $\tau$ be as in Eq. (11). Then,*

$$\tau(s) = \left( I_d \otimes \Phi(s)^\top \right) H_M \left( I_d \otimes \Phi(s) \right) + C, \tag{12}$$

*where $I_d$ is the identity in $\mathbb{R}^{d \times d}$ and $H_M$ is the matrix in $\mathbb{R}^{dk \times dk}$ defined by the entries*

$$\left( H_M \right)_{(i-1)k+h, (j-1)k+z} = \left\langle M(:, i, h), M(:, j, z) \right\rangle, \quad i, j = 1, \dots, d, \quad h, z = 1, \dots, k.$$

The arguments above motivate us to consider the following set of conditioning functions:

$$\mathcal{T}_\Phi = \left\{ \tau(\cdot) = \left( I_d \otimes \Phi(\cdot)^\top \right) H \left( I_d \otimes \Phi(\cdot) \right) + C \ \middle|\ \text{such that } H \in \mathbb{S}_+^{dk}, C \in \mathbb{S}_+^d \right\}. \tag{13}$$

To highlight the dependency of a function $\tau \in \mathcal{T}_\Phi$ with respect to its parameter $H$ and $C$, we will denote $\tau = \tau_{H,C}$. Evidently, $\mathcal{T}_\Phi$ contains the space of all unconditional estimators $\mathcal{T}^{\text{const}}$. We consider $\mathcal{T}_\Phi$ equipped with the canonical norm $\|\tau_{H,C}\|^2 = \|(H,C)\|_F^2 = \|H\|_F^2 + \|C\|_F^2$, where, recall, $\|\cdot\|_F$ denotes the Frobenius norm. The following two standard assumptions will allow us to design and analyse our method.

**Assumption 2.** *The optimal function $\tau_\rho$ belongs to $\mathcal{T}_\Phi$, namely there exist $H_\rho \in \mathbb{S}_+^{dk}$ and $C_\rho \in \mathbb{S}_+^d$, such that $\tau_\rho(\cdot) = \tau_{H_\rho,C_\rho}(\cdot) = \big(I_d \otimes \Phi(\cdot)^\top\big)H_\rho\big(I_d \otimes \Phi(\cdot)\big) + C_\rho$.*

**Assumption 3.** *There exists $K > 0$ such that $\|\Phi(s)\| \leq K$ for any $s \in \mathcal{S}$.*

Asm. 2, known as well-specified setting assumption (see e.g. [33]), is a standard assumption in learning theory and it allows us to restrict the conditional meta-learning problem in Eq. (5) to $\mathcal{T}_\Phi$, rather than to the entire space $\mathcal{T}$ of measurable functions. Asm. 3 ensures that the meta-objective is Lipschitz (see below).

**The convex surrogate problem.** We start from observing that, exploiting the generalization properties of the within-task algorithm (see Prop. 6 in App. A), we can write the following

$$\mathcal{E}_\rho(\tau) \leq \mathbb{E}_{(\mu,s)\sim\rho}\,\mathbb{E}_{Z\sim\mu^n}\,F_Z(\tau(s)), \qquad F_Z(\theta) = \mathcal{R}_{Z,\theta}(A(\theta,Z)) + \frac{2L^2}{n}\mathrm{Tr}\Big(\theta\frac{X^\top X}{n}\Big)$$

where $X \in \mathbb{R}^{n\times d}$ is the matrix with the inputs vectors $(x_i)_{i=1}^n$ as rows. The inequality above suggests us to introduce the surrogate problem

$$\min_{\tau\in\mathcal{T}}\hat{\mathcal{E}}_\rho(\tau), \qquad \hat{\mathcal{E}}_\rho(\tau) = \mathbb{E}_{(\mu,s)\sim\rho}\,\mathbb{E}_{Z\sim\mu^n}\,F_Z(\tau(s)). \tag{14}$$

We stress that the surrogate problem we take here is different from the one considered in previous work [12, 15, 14, 9], where the authors considered as meta-objective only a part of the function above, namely, $\mathbb{E}_{(\mu,s)\sim\rho}\,\mathbb{E}_{Z\sim\mu^n}\big[\mathcal{R}_{Z,\tau(s)}(A(\tau(s),Z))\big]$. As we will see in the following, such a choice is more appropriate for the problem at hand, since, differently from the meta-objective used in previous literature, it will allow us to develop a conditional meta-learning method that is theoretically grounded also for linear representation learning.

Exploiting Asm. 2, the surrogate problem in Eq. (14) can be restricted to the class of linear functions $\mathcal{T}_\Phi$ in Eq. (13) and it can be rewritten more explicitly as

$$\min_{H\in\mathcal{S}^{dk},C\in\mathbb{S}_+^d}\mathbb{E}_{(\mu,s)\sim\rho}\,\mathbb{E}_{Z\sim\mu^n}\,\mathcal{L}\big(H,C,s,Z\big), \qquad \mathcal{L}\big(H,C,s,Z\big) = F_Z\big(\tau_{H,C}(s)\big). \tag{15}$$

In the following proposition we outline some useful properties of the meta-loss $\mathcal{L}\big(\cdot,\cdot,s,Z\big)$ introduced above (such as convexity) supporting its choice as surrogate meta-loss.

**Proposition 4** (Properties of the surrogate meta-loss $\mathcal{L}$)**.** *For any $Z \in \mathcal{D}$ and $s \in \mathcal{S}$, the function $\mathcal{L}\big(\cdot,\cdot,s,Z\big)$ is convex and one of its subgradients is given, for any $H \in \mathbb{S}_+^{dk}$ and $C \in \mathbb{S}_+^d$, by*

$$\nabla\mathcal{L}\big(H,\cdot,s,Z\big)(C) = \hat{\nabla}, \qquad \nabla\mathcal{L}\big(\cdot,C,s,Z\big)(H) = \big(I_d\otimes\Phi(s)\big)\hat{\nabla}\big(I_d\otimes\Phi(s)^\top\big),$$

*where*

$$\hat{\nabla} = -\frac{\lambda}{2}\tau_{H,C}(s)^\dagger w_{\tau_{H,C}(s)} w_{\tau_{H,C}(s)}^\top \tau_{H,C}(s)^\dagger + \frac{2L^2 X^\top X}{n^2}.$$

*Moreover, by Asm. 1 and Asm. 3,*

$$\big\|\nabla\mathcal{L}\big(\cdot,\cdot,s,Z\big)(H,C)\big\|_F \leq (1+K^2)(LR)^2\big(2^{-1}+2n^{-1}\big).$$

The proof of Prop. 4 is reported in App. D.2. It follows from combining results from [15] with the composition of the linear parametrization of the functions $\tau_{H,C} \in \mathcal{T}_\Phi$.

**The conditional Meta-Learning estimator.** The meta-learning strategy we propose consists in applying Stochastic Gradient Descent (SGD) on the surrogate problem in Eq. (15). Such a meta-algorithm is implemented in Alg. 1: we assume to observe a sequence of i.i.d. pairs $(Z_t, s_t)_{t=1}^T$ of training datasets and side information, and at each iteration we update the conditional parameters $(H_t, C_t)$ by performing a step of constant size $\gamma > 0$ in the direction of $-\nabla\mathcal{L}(\cdot,\cdot,s_t,Z_t)(H_t,C_t)$ and a projection step on $\mathbb{S}_+^{dk}\times\mathbb{S}_+^d$. Finally, we output the conditioning function $\tau_{\bar{H},\bar{C}}$ parametrized by $(\bar{H},\bar{C})$, the average across all the iterates $(H_t,C_t)_{t=1}^T$. The theorem below analyzes the generalization properties of such a conditioning function.

**Algorithm 1** Meta-algorithm, SGD on Eq. (15)

---

**Input**  $\gamma > 0$ meta-step size, $H_0 \in \mathbb{S}_+^{dk}$, $C_0 \in \mathbb{S}_+^d$

**Initialization**  $H_1 = H_0 \in \mathbb{S}_+^{dk}$, $C = C_0 \in \mathbb{S}_+^d$

**For**  $t = 1$ to $T$

   Receive $(\mu_t, s_t) \sim \rho$ and $Z_t \sim \mu_t^n$

   Let $\theta_t = (I_d \otimes \Phi(s_t)) H_t (I_d \otimes \Phi(s_t)^\top) + C_t$ and compute $w_{\theta_t} = A(\theta_t, Z_t)$ by Eq. (2)

   Compute $\nabla \mathcal{L}(\cdot, \cdot, s_t, Z_t)(H_t, C_t)$ as in Prop. 4 with $w_{\theta_t}$

   Update $(H_{t+1}, C_{t+1}) = \text{proj}_\Theta \big( (H_t, C_t) - \gamma \nabla \mathcal{L}(\cdot, \cdot, s_t, Z_t)(H_t, C_t) \big)$

**Return**  $\bar{H} = \dfrac{1}{T} \sum_{t=1}^{T} H_t, \bar{C} = \dfrac{1}{T} \sum_{t=1}^{T} C_t$

---

**Theorem 5** (Excess risk bound for the conditioning function returned by Alg. 1)**.** *Let Asm. 1 and Asm. 3 hold. For any $s \sim \rho_\mathcal{S}$, recall the conditional covariance matrices $W(s)$ and $C(s)$ introduced in Thm. 1. Let $\tau_{H,C}$ be a fixed function in $\mathcal{T}_\Phi$ such that $\text{Ran}(W(s)) \subseteq \text{Ran}(\tau_{H,C}(s))$ for any $s \sim \rho_\mathcal{S}$. Let $\bar{H}$ and $\bar{C}$ be the outputs of Alg. 1 applied to a sequence $(Z_t, s_t)_{t=1}^T$ of i.i.d. pairs sampled from $\rho$ with an appropriate meta-step size $\gamma$. Then, in expectation with respect to the sampling of $(Z_t, s_t)_{t=1}^T$,*

$$\mathbb{E}\, \mathcal{E}_\rho(\tau_{\bar{H},\bar{C}}) - \mathcal{E}_\rho^* \leq \frac{\mathbb{E}_{s \sim \rho_\mathcal{S}} \text{Tr}\big( \tau_{H,C}(s)^\dagger W(s) \big)}{2} + \frac{2L^2 \mathbb{E}_{s \sim \rho_\mathcal{S}} \text{Tr}\big( \tau_{H,C}(s) C(s) \big)}{n}$$
$$+ \Big( \frac{1}{2} + \frac{2}{n} \Big) \frac{(1 + K^2)(LR)^2 \, \|(H - H_0, C - C_0)\|_F}{\sqrt{T}}.$$

*Proof (Sketch).* The detailed proof is reported in App. D.4. Exploiting the fact that, for any $\tau \in \mathcal{T}$, $\mathcal{E}_\rho(\tau) \leq \hat{\mathcal{E}}_\rho(\tau)$ and adding $\pm \hat{\mathcal{E}}_\rho(\tau_{H,C})$, we can write the following

$$\mathbb{E}_\mathbf{Z}\, \mathcal{E}_\rho(\tau_{\bar{H},\bar{C}}) - \mathcal{E}_\rho^* \leq \text{A}(\tau_{H,C}) + \text{B}(\tau_{H,C})$$
$$\text{A}(\tau_{H,C}) = \mathbb{E}_\mathbf{Z}\, \hat{\mathcal{E}}_\rho(\tau_{\bar{H},\bar{C}}) - \hat{\mathcal{E}}_\rho(\tau_{H,C}) \qquad \text{B}(\tau_{H,C}) = \hat{\mathcal{E}}_\rho(\tau_{H,C}) - \mathcal{E}_\rho^*. \tag{16}$$

The term $\text{A}(\tau_{H,C})$ can be controlled according to the convergence properties of the meta-algorithm in Alg. 1 as described in Prop. 12. Regarding the term $\text{B}(\tau_{H,C})$, exploiting the definition of the within-task algorithm in Eq. (2) as minimum, for any $\tau \in \mathcal{T}$ such that $\text{Ran}(\mathbb{E}_{\mu \sim \rho(\cdot|s)} w_\mu w_\mu^\top) \subseteq \text{Ran}(\tau(s))$ for any $s \sim \rho_\mathcal{S}$, we can rewrite

$$\text{B}(\tau) \leq \frac{\mathbb{E}_{(\mu,s) \sim \rho} \text{Tr}\big( \tau(s)^\dagger w_\mu w_\mu^\top \big)}{2} + \frac{2L^2 \mathbb{E}_{(\mu,s) \sim \rho} \text{Tr}\big( \tau(s) \mathbb{E}_{x \sim \eta_\mu} xx^\top \big)}{n}.$$

The desired statement then derives from combining the two parts above and optimizing with respect to $\gamma$. $\qquad \square$

**Remark 3** (Online variant of Eq. (2))**.** *Similarly to the bias regularization framework in [14], for the online inner family in Rem. 2, we approximate the meta-subgradient in Prop. 4 by replacing the batch minimizer $A(\tau_{H,C}(s), Z)$ in Eq. (2) with the last iterate of the online algorithm in Eq. (3).*

**Proposed vs. optimal conditioning function.** Specializing the bound in Thm. 5 to the best conditioning function $\tau_\rho$ in Prop. 2, thanks to Asm. 2, we get, up to contants, the following bound for our estimator,

$$\mathbb{E}_{s \sim \rho_\mathcal{S}} \big\| W(s)^{1/2} C(s)^{1/2} \big\|_* \, n^{-1/2} + \|(H_\rho - H_0, C_\rho - C_0)\|_F \, T^{-1/2}.$$

From such a bound, we can state that our proposed meta-algorithm achieves comparable performance to the best conditioning function $\tau_\rho$ in hindsight, when the number of observed tasks is sufficiently large. Moreover, recalling the unconditional oracle $\hat{\theta}_\rho$ in Eq. (10) used in previous literature, regarding the second term vanishing with $T$, we observe that our conditional meta-learning approach incurs a cost of $\|(H_\rho - H_0, C_\rho - C_0)\|_F T^{-1/2}$ as opposed to the cost of $\|\hat{\theta}_\rho - \theta_0\| T^{-1/4}$ associated to state-of-the-art unconditional meta-learning approaches (see [15, 5, 22, 9]). Thus, our conditional approach presents a faster convergence rate with respect to $T$ than such unconditional methods, but a complexity term that is expected to be larger due to the larger complexity of the class of functions we

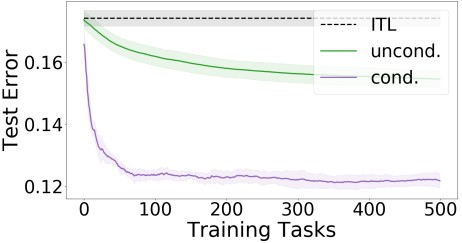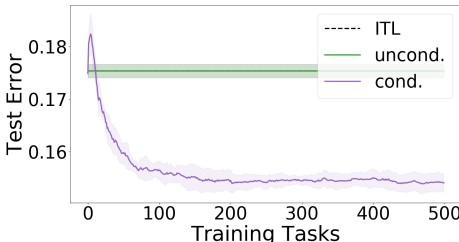

Figure 1: Mean test error (5 generations) on synthetic data. 2 (Left) and 6 (Right) clusters.

are working with. Such a faster rate with respect to $T$ is essentially due to our formulation of the problem on the entire set of positive-semidefinite matrices (with no trace constraints). This in fact allows us to incorporate the within-task regularization parameter $\lambda$ directly in the linear representation and to gain a $\sqrt{T}$ order that was lost in previous literature when tuning with respect to the parameter $\lambda$. At the same time, this allows us to develop also a method requiring to tune just one hyper-parameter, while previous unconditional approaches requires to tune two hyper-parameters.

**Comparison to unconditional Meta-Learning.** Specializing Thm. 5 to the best unconditional estimator $\tau_{H,C} \equiv \theta_\rho$ we introduced in Eq. (9), the bound for our estimator becomes, up to constants,

$$\left\| W_\rho^{1/2} C_\rho^{1/2} \right\|_* n^{-1/2} + \left\| \theta_\rho - C_0 \right\| T^{-1/2}.$$

From the bound above, we can conclude that the conditional approach provides, at least, the same guarantees as its unconditional counterpart. Moreover, we stress again that the bound above presents a faster rate with respect to $T$ in comparison to the state-of-the-art unconditional methods.

## 5    Experiments

We now present preliminary experiments in which we compare the proposed conditional meta-learning approach in Alg. 1 (cond.) with the unconditional counterpart (uncond.) and solving the tasks independently (ITL, namely, running the inner algorithm separately across the tasks with the constant linear representation $\theta = I_d \in \mathbb{S}_+^d$). We considered regression problems and we evaluated the errors by $\ell$ the absolute loss. We implemented the online variant of the within-task algorithm introduced in Eq. (3). The hyper-parameter $\gamma$ was chosen by (meta-)cross validation on separate $T_{\mathrm{tr}}$, $T_{\mathrm{va}}$ and $T_{\mathrm{te}}$ respectively meta-train, -validation and -test sets. Each task is provided with a training dataset $Z_{\mathrm{tr}}$ of $n_{\mathrm{tr}}$ points and a test dataset $Z_{\mathrm{te}}$ of $n_{\mathrm{te}}$ points used to evaluate the performance of the within-tasks algorithm. In App. E we report the details of this process in our experiments.

**Synthetic clusters.** We considered two variants of the setting described in Ex. 1 with side information corresponding to the training datasets $Z_{\mathrm{tr}}$ associated to each task. In both settings, we sampled $T_{\mathrm{tot}} = 900$ tasks from a uniform mixture of $m$ clusters. For each task $\mu$, we generated the target vector $w_\mu \in \mathbb{R}^d$ with $d = 20$ as $w_\mu = P(j_\mu)\tilde{w}_\mu$, where, $j_\mu \in \{1, \dots, m\}$ denotes the cluster from which the task $\mu$ was sampled and with the components of $\tilde{w}_\mu \in \mathbb{R}^{d/(10)}$ sampled from the Gaussian distribution $\mathcal{G}(0,1)$ and then $\tilde{w}_\mu$ normalized to have unit norm, with $P(j_\mu) \in \mathbb{R}^{d \times d/(10)}$ a matrix with orthonormal columns. We then generated the corresponding dataset $(x_i, y_i)_{i=1}^{n_{\mathrm{tot}}}$ with $n_{\mathrm{tot}} = 80$ according to the linear equation $y = \langle x, w_\mu \rangle + \epsilon$, with $x$ sampled uniformly on the unit sphere in $\mathbb{R}^d$ and $\epsilon$ sampled from a Gaussian distribution, $\epsilon \sim \mathcal{G}(0, 0.1)$. In this setting, the operator norm of the inputs' covariance matrix is small (equal to $1/d$) and the weight vectors' covariance matrix of each single cluster is low-rank (its rank is $d/(10) = 2$). We implemented our conditional method using the feature map $\Phi : \mathcal{D} \to \mathbb{R}^{2d}$ defined by $\Phi(Z) = \frac{1}{n_{\mathrm{tr}}} \sum_{i=1}^{n_{\mathrm{tr}}} \phi(z_i)$, with $\phi(z_i) = \mathrm{vec}(x_i(y_i, 1)^\top)$, where, for any matrix $A = [a_1, a_2] \in \mathbb{R}^{d \times 2}$ with columns $a_1, a_2 \in \mathbb{R}^d$, $\mathrm{vec}(A) = (a_1, a_2)^\top \in \mathbb{R}^{2d}$. In Fig. 1, we report the results we got on an environment of tasks generated as above with $m = 2$ (Left) and $m = 6$ (Right) clusters, respectively. As we can see, when the clusters are two, the unconditional approach outperforms ITL (as predicted from previous literature), but the unconditional method is in turn outperformed by our conditional counterpart. When the number of clusters raises to six, the performance of unconditional meta-learning degrades to the same performance of ITL,

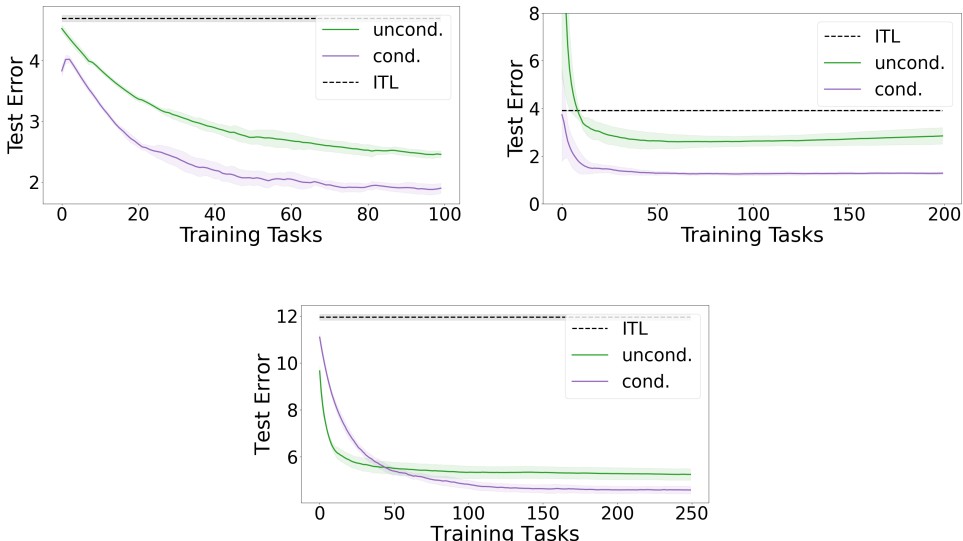

Figure 2: Mean test error (5 splits) on Lenk (Top-Left), Movielens-100k (Top-Right), Jester-1 (Bottom) dataset.

while conditional meta-learning outperforms both methods. Summarizing, the more the heterogeneity of the environment (number of clusters) is significant, the more the conditional approach brings advantage with respect to the unconditional one. This is in line with Ex. 1.

**Real datasets.** We tested the performance of the methods also on the regression problem on the computer survey data from [23] (see also [28]). $T_{\mathrm{tot}} = 180$ people (tasks) rated the likelihood of purchasing one of $n_{\mathrm{tot}} = 20$ computers. The input represents $d = 13$ computers' characteristics and the label is a rate in $\{0, \ldots, 10\}$. In this case, we used as side information the training datapoints $Z = (z_i)_{i=1}^{n_{\mathrm{tr}}}$ and the feature map $\Phi : \mathcal{D} \to \mathbb{R}^{d+1}$ defined by $\Phi(Z) = w_Z$, with $w_Z$ the solution of Tikhonov regularization with the squared loss, namely, the vector satisfying $(\hat{X}^\top \hat{X} + I_{d+1})w_Z = \hat{X}^\top y$, where, $\hat{X} \in \mathbb{R}^{(d+1) \times n}$ is the matrix obtained by adding to the matrix $X \in \mathbb{R}^{n \times d}$ one column of ones at the end. Fig. 2 (Top-Left) shows that also in this case, the unconditional approach outperforms ITL, but the performance of its conditional counterpart is much better.

We also tested the performance of the methods on the Movielens-100k and Jester-1 real-world datasets, containing ratings of users (tasks) to movies and jokes (points), respectively. Recommendation system settings with $d$ items can be interpreted within the meta-learning setting by considering each data point $(x, y)$ to have input $x \in \mathbb{R}^d$ to be the one-hot encoding of the current item to be rated (e.g. a movie or a joke) and $y \in \mathbb{R}$ the corresponding score, see e.g. [12] for more details. We restricted the original dataset to the $n_{\mathrm{tot}} = 20$ most voted movies/jokes (as a consequence, by formulation, $d = 20$). We guaranteed each user voted at least 5 movies/jokes, which led to a total of $T_{\mathrm{tot}} = 400/450$ tasks (i.e. users). In both cases, we used as side information the training datapoints $Z = (z_i)_{i=1}^{n_{\mathrm{tr}}}$. For the Movielens-100k dataset we used the same feature map described for the synthetic clusters experiments in Fig. 1. For the Jester-1 dataset, let $M$ and $m$ denote the maximum and minimum rating value that can be assigned to a joke. We adopted the feature map $\Phi : \mathcal{D} \to \mathbb{R}^{2d+1}$ such that, for any dataset $Z = (x_i, y_i)_{i=1}^n$, we have $\Phi(Z) = \big(\mathrm{vec}(\tilde{\Phi}(Z)); 1\big)$, where vec denotes the vectorization operator (i.e. mapping a matrix in the vector concatenating all its columns) and $\tilde{\Phi} : Z \to \mathbb{R}^{d \times 2}$ is such that $\tilde{\Phi}(Z) = \big(\cos(\alpha(Z)), \sin(\alpha(Z))\big) \odot (\sum_{i=1}^n x_i)$ with $\alpha(Z) = \sum_{i=1}^n x_i \left(\frac{\pi}{4} \frac{M - y_i}{M - m}\right)$ and $\odot$ denoting the Hadamard (entry-wise) product broad-casted across both columns. The rationale behind this feature map is to represent as similar vectors those users with similar scores for the same movies. In particular, each item-score pair observed in training is represented as a unitary vector in $\mathbb{R}_{++}^2$, with the angle depending on the score attributed to that item (the vector corresponds to zero if that movie was not observed at the training time). As it can be noticed in Fig. 2 (Top Right and Bottom), the

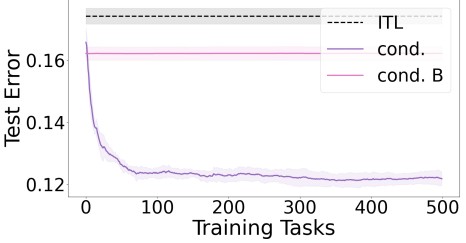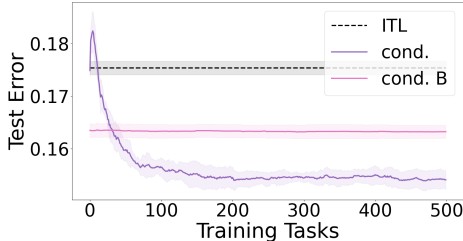

Figure 3: Mean test error (5 generations) of our method and [14] on 2 (Left) and 6 (Right) clusters.

proposed approach performs significantly better than ITL and its unconditional counterpart also on these two benchmarks. This suggests that groups of users might rely each on similar features (but different from those of other groups) to rate an item in the dataset (respectively a movie or a joke).

**Comparison with [14].** We conclude the experimental section by comparing the performance of our method with the conditional meta-learning approach for biased regularization proposed in [14]. In that case, the tasks' target vectors are assumed to be all close to a common bias vector rather than sharing the same low-dimensional linear representation, as instead in our method. The representation setting is known to be a more relevant and effective framework in many scenarios in comparison to the bias one (see e.g. [28]). This is confirmed in Fig. 3 where our conditional representation learning method ('cond.') significantly outperforms the one in [14] ('cond. B') in the synthetic settings used in Fig. 1, since the tasks' similarity leveraged by [14] is not appropriate in these cases.

## 6   Conclusion

We proposed a conditional meta-learning approach aiming at learning a function mapping task's side information into a linear representation that is well suited for the task at hand. We theoretically and experimentally showed that the proposed conditional approach is advantageous with respect to the standard unconditional counterpart when the observed tasks share heterogeneous linear representations. As a consequence of our analysis we also developed a new unconditional meta-learning variant requiring tuning less hyper-parameters and relying on faster rates with respect to state-of-the-art unconditional approaches. We identify two future directions addressing the limitations of our method. A first question left opened is how to design a suitable feature map $\Phi$ when the tasks' training data is used as side information. Following [32, 37], we adopted a mean embedding representation. However, given the importance played by such feature map in Thm. 5, it will be worth investigating better alternatives. Secondly, it will be valuable to investigate how to predict non-linear conditioning functions (similarly to e.g. [7, 16, 32]) and use less expensive algorithms to update the positive matrices, such as the Frank-Wolfe algorithm used in [9] for unconditional settings. In this last case, according to our analysis, applying standard convergence rates for Frank-Wolfe algorithm, we expect the meta-learning algorithm based on Frank-Wolfe iteration to incur a slower rate of order $T^{-1/4}$ (instead of $T^{-1/2}$ for the SGD method proposed here) in Thm. 5, paying the computational benefits in terms of statistical performance.

## Acknowledgments and Disclosure of Funding

This work was supported in part by SAP SE and EPSRC Grant N. EP/P009069/1. C.C. acknowledges the support of the Royal Society (grant SPREM RGS\R1\201149) and Amazon.com Inc. (Amazon Research Award – ARA). G.D. acknowledges Leonardo SpA for funding her participation to the conference.

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
