# OpenReview forum: "Conditional Meta-Learning of Linear Representations"
_NeurIPS.cc/2022/Conference — NeurIPS 2022 Accept_

### Official Review · Reviewer_4JRE · 2022-07-09

**Rating:** 7
**Confidence:** 2
**Soundness:** 2 fair
**Presentation:** 3 good
**Contribution:** 2 fair

**Summary:**

This paper addresses representation learning in meta learning setting when tasks' distribution cannot be captured by a single cluster and representation. It proposes to group tasks into different clusters where  tasks in the same cluster share the same representation, and to map task side information to a task specific linear representation via conditional function. The paper provides theoretical and experimental demonstration of the advantages of the proposed approach over standard meta-learning approach.


**Questions:**

What's the differences and advantages of the proposed method comparing with [14] which is also a conditional meta learning method?


**Limitations:**

The authors addressed the limitations explicitly, and did not discuss potential negative societal impact of the work.


**Strengths And Weaknesses:**

Strengths:
+ The paper proposes a conditional meta-learning approach to deal with representation learning among hyterogenous tasks.

+ It derives a new unconditional meta-learning variant that can have faster learning rate and requires less hyper-parameter tuning.

Weaknesses:
- The algorithm for updating the positive matrices is computationally expensive.

- Currently the conditional function is linear, it cannot deal with non-linear mapping from task side information to a representation.

---

> ### Author Response · Authors · 2022-08-01
> **Reply to R4JRE**
>
> 1) __R.__ *The algorithm for updating the positive matrices is computationally expensive.* __A.__ We agree with the reviewer regarding the computational considerations. However, we would like to point out that, as we already mentioned in our conclusion, it is possible to employ significantly less expensive approximation strategies to perform the update step, such as methods based on Frank-Wolfe (FW) iterations (se e.g. [9]). We did not include this strategy in our paper because of the following reasons. __a)__ The main goal of our paper is to characterize which meta-learning regimes can benefit from learning a shared representation and show that it is possible to define an algorithm able to attain such an advantage. In this sense, our goal was more towards models and theory rather than pure efficiency, which can be investigated using approximate methods such as the Frank-Wolfe iterations mentioned above. __b)__ According to our analysis, applying standard convergence rates for FW algorithm, the meta-learning algorithm based on Frank-Wolfe iteration would incur a slower rate of order $T^{-\frac{1}{4}}$ (instead of our $T^{-\frac{1}{2}}$) in Thm. 5. Hence, it would pay the computational benefits in terms of statistical performance. We will expand our discussion in the conclusion to better clarify this point.
>
>
> 2) __R.__ *About the linearity of the conditional function.* __A.__ Please refer to our replies 2) to reviewer oWcc and 1) to reviewer 1CMy.
>
>
> 3) __R.__ *About the differences and advantages w.r.t. [14].* __A.__ Please refer to our replies 1) and 4) to reviewer 5r8e.

---

### Official Review · Reviewer_1CMy · 2022-07-11

**Rating:** 7
**Confidence:** 2
**Soundness:** 3 good
**Presentation:** 3 good
**Contribution:** 3 good

**Summary:**

This paper presents a principled algorithm for performing meta-learning conditioned on available side information in the linear setting. The paper theoretically justifies that conditional meta-learning can produce superior convergence rates over unconditioned counter-part in the particular problem framing described.
They conduct experiments on both synthetic and real data settings to show the efficacy of their results

**Questions:**

Relevant related work missing :
Can the authors comment about the connection of their approach to the following - (when considered in the linear setting)

[1] Task Similarity Aware Meta-Learning: https://proceedings.mlr.press/v161/zhou21a/zhou21a.pdf


**Nit**
* Figure 1 - the caption should be (left) and (right) - no ?

**Limitations:**

The authors rightly identify multiple limitations of their work and suggest to tackle them in future work

**Strengths And Weaknesses:**

**Strengths**
1. This paper tackles an important problem of conditioned meta-learning when side information is provided. They prove faster convergence rates for a  flavor of the conditional setting.
2. The paper is well written and intuitively structured. The chosen notation for the proofs make them a bit easier to go through.
3. Experiments, though simple, drive the point home about the relevance of the setting explored.


**Weaknesses**
1. The paper presents a core assumption (2), that the optimal function $\tau_{\rho}$ lies in the given space but provide no intuition or at least rough justification for why this could be the case. Though the experiments suggest that this does not prevent good performance over baselines, it is unclear that this is not a quirk of the setting they test in.
2. k is introduced in section 4 as the dimension of the output feature map given side information but subsequently it is not discussed how it is set in experiments - did I miss this ?
3. The experiments seem a bit limited - but this seems defensible in light of the extensive theory provided. Eg - an exploration of a wider range of feature mapping functions for the side information would be a good ablation to perform.

---

> ### Author Response · Authors · 2022-08-01
> **Reply to R1CMy**
>
> 1) __R.__ *About the assumption for the optimal conditioning function to belong to the hypotheses space.* __A.__ We did not comment upon this since this is a standard assumption in learning theory, known as a well-specified setting (see e.g. [33]). In ill-specified settings, we can either have an additional so-called irreducible error (representing the bias introduced by choosing an hypotheses space) or an approximation error that can be controlled by model selection if the reproducing kernel used is universal (see [34]). In the latter case this might still yield universal consistency (i.e. the excess risk going to zero as the number of meta-training tasks increases) but with slower rates than $1/\sqrt{T}$ depending on how “close” the ideal function is to the hypotheses space. We will comment and add the related pointers to the literature regarding this assumption.
>
>
> 2) __R.__ *About the setting of the dimension k of the output feature map.* __A.__ The dimension k depends on the feature map we use, see e.g. lines 291-292, 305-307, 320-324. Note that our algorithm could be extended to settings where the feature map takes values in an infinite dimensional feature space (e.g. when using a Gaussian kernel) using a similar strategy to the “kernel trick” but at the tasks level. For the sake of simplicity we focused our discussion only to finite dimensional feature spaces but we will clarify this in the text.
>
>
> 3) __R.__ *About an exploration of a wider range of feature mapping functions in the experiments.* __A.__ In our experiments we tried to use different feature maps, but we decided to report only the performance we got with the best one. We think that adding other feature maps in the plots would only add noise to the figures, without adding additional contribution supporting the main message of the paper: the significant advantage of our conditional approach in comparison to the unconditional counterpart for linear representation learning.
>
>
> 4) __R.__ *Connection to ref. [Task Similarity Aware Meta-Learning].* __A.__ We thank the reviewer for pointing out this additional reference. Differently from our work, the suggested paper uses a method combining a k-means strategy to the standard MAML method. Since MAML considers meta-parameters that are starting points of a bias vector, the suggested paper is more related to [14] than to our setting, where the meta-parameters correspond to linear representations. In addition, although the suggested paper is focused on theory, the authors do not manage to provide a complete excess risk bound analysis of the learning behavior of their estimator. This is mostly due to the lack of convexity in the k-mean steps. As a matter of fact, the conclusion below their Cor. 1 reads as ‘Cor. 1 shows that if the learner can assign the task T into a correct group, the method would be expected to have smaller expected excess risk than standard MAML’. However, the authors are not able to theoretically show whether or under what hypotheses the method satisfies this assumption. In contrast, in our work we manage to provide a complete excess risk bound analysis for the proposed method. This is possible thanks to the fact that our method is able to perform a “smooth” (and implicit) clustering of the tasks by learning the conditioning function mapping task’s side information to a tasks’ specific linear representation. We will add the recommended paper and a discussion about it.
>
>
> 5) __R.__ *About the caption in Fig. 1.* __A.__ The reviewer is correct, thanks. We will amend it.

---

### Official Review · Reviewer_oWcc · 2022-07-13

**Rating:** 5
**Confidence:** 3
**Soundness:** 3 good
**Presentation:** 3 good
**Contribution:** 3 good

**Summary:**

This paper introduces a new representation learning method designed for working with a diverse set of heterogeneous tasks. In contrast to prior works that learn task vectors (via a conditional mapping that takes in side information) that are assumed to be close to a common bias vector, this work directly learns linear representations for the task at hand. They provide theoretical analyses for this approach and validate the method on small datasets.

**Questions:**

- The proposed method appears to be limited to learned linear representations that will be adapted to the tasks at hand, so it's a bit tricky for me to see how applicable/scalable this approach is to more complex, real-world datasets. The paper also compared their approach to very weak baselines (e.g. the unconditional meta-learning formulation). Would the authors comment on this point?

**Limitations:**

The authors addressed a few limitations of their approach in the conclusion, but did not provide any details on potential negative societal impacts of their work.

**Strengths And Weaknesses:**

Strengths:
- The paper provides rigorous theoretical analyses regarding their approach (e.g. excess risk, conditions for optimality, relationship to existing approaches, etc.).

Weaknesses:
- The empirical evaluations are quite weak -- I've elaborated a bit more on this in the "Questions" section.
- This is a very minor point, but there are various places throughout the paper where the writing could be cleaned up to make the presentation more clear (e.g. "w.r.t. the bias one in l. 36).

---

> ### Author Response · Authors · 2022-08-01
> **Reply to RoWcc**
>
> 1) __R.__ *About places where the writing could be cleaned up (e.g. "w.r.t. the bias one in l. 36).* __A.__ We thank the reviewer and will polish the text according to their suggestions.
>
>
> 2) __R.__ *About the possible limitations of using linear representations.* __A.__ We agree with the reviewer that non-linear meta-representation learning is an interesting field. However, we would like to point out that, there is a large body of work focusing on linear models, e.g. when working with tabular data. In particular, we have cited a few papers where linear models, possibly integrated with handcrafted or pre-trained input feature maps, are very effective in real-world applications (see for instance [3,5,9,12,15,22]). In this sense, the focus of our paper is not a limitation, but rather devoting attention to these specific settings. Additionally, note that our algorithm can be applied also to learning non-linear meta-representations. However, our theoretical guarantees should be consequently adapted in order to address the lack of convexity of the overall meta-learning problem.
>
>
> 3) __R.__ *The paper also compared their approach to very weak baselines (e.g. the unconditional meta-learning formulation).* __A.__  Our experiments compare the proposed conditional meta-learning method with its unconditional counterpart from [15]. The latter is considered state-of-the-art in the context of linear meta-representation learning, therefore we don’t understand why the reviewer is referring to it as a weak baseline. The method in [15] is the closest unconditional counterpart / benchmark to our method, as a consequence, we decided to compare with it and on the datasets used in that paper.

---

### Official Review · Reviewer_5r8e · 2022-07-13

**Rating:** 4
**Confidence:** 3
**Soundness:** 4 excellent
**Presentation:** 3 good
**Contribution:** 2 fair

**Summary:**

Meta-learning, but additionally using “side information” to resolve contexts among a heterogeneous collection of tasks, in order to induce better task-specific representations. The side information is captured through a linear representation, which is then relaxed suitably for tractable optimization. As a special case, the formulation also reduces one hyper-parameter choice in the unconditional case. The paper then demonstrates results using experiments with a synthetic dataset (of clusters), and a few real-world datasets.

**Questions:**

- (Question) I wonder whether it might be possible to add to the text an explanation intuitively motivating the sqrt{m} factor in Example 1.

- (Suggestion) It would be helpful to contrast Example 1 with what would happen if the method from ref. [14] were to be applied (to better elucidate the significance of this new approach).

- (Suggestion) It might be helpful for readers if the paper could interpret the changed modeling assumptions (and notion of task similarity), and elaborate on the general nature of problems (incl. example scenarios) where the approach in this paper could lead to substantial improvements over the method presented in ref. [14].


**Limitations:**

Yes, the authors seem to have adequately addressed the limitations (as avenues for future work) and potential social impact (this work is abstracted to substantial generality beyond any specific application).

**Strengths And Weaknesses:**

The paper follows somewhat closely the presentation in ref. [14], building on it with what might be interpreted as a modeling improvement that also corresponds to a different notion of task similarity. The claims are augmented with what appears to be a rigorous characterization. Though I have not carefully verified the math, the results seem reasonable & understandable. My main concern regards the significance of the new result, given what has previously been elucidated in ref. [14].

[14]: The Advantage of Conditional Meta-Learning for Biased Regularization and Fine Tuning, NeurIPS 2020

---

> ### Author Response · Authors · 2022-08-01
> **Reply to R 5r8e**
>
> 1) __R.__ *About the significance of the new results, considering ref. [14].*  __A.__ We would like to stress that the similarity with [14] __concerns only the structure of the two papers but not their contents and significance__ (see for instance our reply 4 below regarding Ex. 1). In particular, we want to stress that our main contribution is to characterize for the first time the benefits of conditional vs unconditional meta-representation learning. This is a significantly different setting from the meta-bias learning setting in [14]. Conditional meta-representation learning, even for linear settings, was still an open question in the literature (see e.g. [3,5,9,12,14,15,22] and references therein)  and thus our paper is of significance for the community. Moreover, the analysis and the algorithm of the present paper differ quite significantly to [14] due to a new formulation of the problem, a different meta-objective and a different interpretation of the results. Our results are non trivial and cannot be derived from [14]. We found presenting our paper with a similar structure to [14] would help appreciate our contributions rather than making them less clear. We thank the reviewer for raising the point and we will further clarify the relation with [14] throughout the text.
>
>
> 2) __R.__ *Providing an intuition for the $\sqrt{m}$ factor in Ex. 1.* __A.__ The gain factor follows from the fact that the weight vectors $w_\mu$ sampled from the different clusters share disjoint supports (they have orthogonal representations). This allows us to rewrite the overall clusters weight vectors’ covariance as the average of the intra clusters weight vectors’ covariances. The $\sqrt{m}$ term comes from this rewriting and the quadratic behavior of the covariance matrix. More details about this can be found in the appendix. We will explain this in the main body.
>
>
> 3) __R.__ *Considering Ex. 1 to the method in ref. [14].* __A.__ That is a good point. We recall that the best performance in [14] can be controlled by the weight vectors’ variance (Exp_s || $w_\mu$ - Exp_{$\mu$ |s} w_$\mu$ ||$^2$)$^{1/2}$ $n^{-1/2}$. In the setting of Ex. 1, such a term can be arbitrary large without adding further assumptions about the mean of the conditioned distribution $\mu|s$, namely the term Exp_{$\mu$ |s} $w_\mu$. Thus, in the setting of Ex. 1, [14] cannot guarantee any advantage between conditional, unconditional or independent learning. This is due to the fact that the two frameworks benefit from different assumptions on the meta-distribution that are orthogonal to each other. We thank the reviewer for the suggestion, we will clarify the distinction between the present paper and [14].
>
>
> 4) __R.__ *Regarding modeling assumptions.* __A.__ The focus of [14] is on meta-learning the parameters for biased regularization algorithms. In these settings the individual task’s solution vectors are assumed to be close to a bias vector. In contrast, in this work we follow ideas from the multi-task linear representation learning literature, where we assume that tasks share a low-dimensional representation (see e.g. [27, 25, 15, 24, 35, 5, 22, 9]). We commented on the relation with [14] from the theoretical, modeling and empirical perspectives in lines 32-39, 137-139, 209-214, 332-339 and Fig. 3. From such comparisons, we concluded that the use of a representation framework instead of a bias one is more appropriate when the tasks share a common representation-valued function instead of a bias vector-valued one.

---

> > ### Comment · Reviewer_5r8e · 2022-08-09
> > **Thank you for the thorough response**
> >
> > I appreciate the clarifications and thorough response, and now better understand the contrast with ref. [14]. I shall go over the paper once again in light of this new understanding, and reconsider my rating.

---

> > > ### Author Response · Authors · 2022-08-09
> > > **Reply to Reviewer 5r8e**
> > >
> > > Thank you for your reply and time.

---

### Meta-Review · Area_Chair_rtpv · 2022-08-25

**Recommendation:** Accept
**Confidence:** Less certain

**Metareview:**

This work presents a new meta-learning algorithm to infer linear representation of task side information. It introduces modelling improvement over existing conditional meta-learning works with shared solution vectors, conducts rigorous theoretical analysis, and shows improvement performance with preliminary experiments. Its unconditional meta-learning variant has faster learning rate and requires less hyper-parameter tuning than SOTA methods.

The reviewers had concerns in the original reviews including the similarity to existing work [14], weak empirical evaluation, computation complexity, and limited linear representation. The authors' feedback addressed / should have address most concerns, and multiple reviewers increased their rating. I would encourage the authors to incorporate their feedback into the revision.

**Award:**

No

---

### Decision · Program_Chairs · 2022-09-14

Accept